# Oxidized Mitochondrial DNA Engages TLR9 to Activate the NLRP3 Inflammasome in Myelodysplastic Syndromes

**DOI:** 10.3390/ijms24043896

**Published:** 2023-02-15

**Authors:** Grace A. Ward, Robert P. Dalton, Benjamin S. Meyer, Amy F. McLemore, Amy L. Aldrich, Nghi B. Lam, Alexis H. Onimus, Nicole D. Vincelette, Thu Le Trinh, Xianghong Chen, Alexandra R. Calescibetta, Sean M. Christiansen, Hsin-An Hou, Joseph O. Johnson, Kenneth L. Wright, Eric Padron, Erika A. Eksioglu, Alan F. List

**Affiliations:** 1Cancer Biology PhD Program, University of South Florida and H. Lee Moffitt Cancer Center & Research Institute, Tampa, FL 33612, USA; 2Department of Malignant Hematology, H. Lee Moffitt Cancer Center & Research Institute, Tampa, FL 33612, USA; 3Department of Immunology, H. Lee Moffitt Cancer Center & Research Institute, Tampa, FL 33612, USA; 4Division of Hematology, Department of Internal Medicine, National Taiwan University Hospital Taipei, Taipei 100229, Taiwan; 5Analytic Microscopy Core Facility, H. Lee Moffitt Cancer Center & Research Institute, Tampa, FL 33612, USA; 6Precision BioSciences, Inc., Durham, NC 27701, USA

**Keywords:** oxidized mitochondrial DNA, DAMP, Myelodysplastic Syndromes, inflammasome, pyroptosis, Toll-like receptor, TLR9, hematopoiesis, MyD88

## Abstract

Myelodysplastic Syndromes (MDSs) are bone marrow (BM) failure malignancies characterized by constitutive innate immune activation, including NLRP3 inflammasome driven pyroptotic cell death. We recently reported that the danger-associated molecular pattern (DAMP) oxidized mitochondrial DNA (ox-mtDNA) is diagnostically increased in MDS plasma although the functional consequences remain poorly defined. We hypothesized that ox-mtDNA is released into the cytosol, upon NLRP3 inflammasome pyroptotic lysis, where it propagates and further enhances the inflammatory cell death feed-forward loop onto healthy tissues. This activation can be mediated via ox-mtDNA engagement of Toll-like receptor 9 (TLR9), an endosomal DNA sensing pattern recognition receptor known to prime and activate the inflammasome propagating the IFN-induced inflammatory response in neighboring healthy hematopoietic stem and progenitor cells (HSPCs), which presents a potentially targetable axis for the reduction in inflammasome activation in MDS. We found that extracellular ox-mtDNA activates the TLR9-MyD88-inflammasome pathway, demonstrated by increased lysosome formation, IRF7 translocation, and interferon-stimulated gene (ISG) production. Extracellular ox-mtDNA also induces TLR9 redistribution in MDS HSPCs to the cell surface. The effects on NLRP3 inflammasome activation were validated by blocking TLR9 activation via chemical inhibition and CRISPR knockout, demonstrating that TLR9 was necessary for ox-mtDNA-mediated inflammasome activation. Conversely, lentiviral overexpression of TLR9 sensitized cells to ox-mtDNA. Lastly, inhibiting TLR9 restored hematopoietic colony formation in MDS BM. We conclude that MDS HSPCs are primed for inflammasome activation via ox-mtDNA released by pyroptotic cells. Blocking the TLR9/ox-mtDNA axis may prove to be a novel therapeutic strategy for MDS.

## 1. Introduction

Myelodysplastic Syndromes (MDSs) are bone marrow (BM) failure diseases typified by chronic BM inflammation, ineffective hematopoiesis, and peripheral blood (PB) cytopenias [1,2,3]. We and others demonstrated that the danger-associated molecular pattern (DAMP) protein S100A9 plays a critical role in the pathogenesis of MDS by creating an inflammatory microenvironment [4,5,6,7]. S100A9 engages Toll-like receptor (TLR)-4 to initiate pyroptosis in hematopoietic stem and progenitor cells (HSPCs) through the Nod-like receptor 3 (NLRP3) inflammasome complex, leading to the induction of clinically evident ineffective hematopoiesis [5]. Constitutive inflammasome activation prevents HPSCs from differentiating, causing cytopenias and contributing to expansion of the malignant clone. Upon pyroptosis execution, cells expel their intracellular contents, including DAMPs, into the extracellular space triggering a feed-forward process that propagates inflammasome and innate immune activation to neighboring cells [8]. Elucidating those signals is important for understanding why the inflammasome is active in MDS.

Increased cell-free DNA levels have been reported in chronic inflammatory disorders [9,10], where there is also increased reactive oxygen species (ROS) induced during pyroptotic events, which not only further contribute to inflammasome activation but to the release of cell-free DAMPs into the extracellular space [11,12,13,14,15]. Among the DAMPs released by pyroptotic cytolysis, and mitochondrial membrane depolarization, is ROS-oxidized mitochondrial DNA (ox-mtDNA) which is then recognized by pattern recognition receptors (PRRs) [16,17]. Ox-mtDNA can amplify pyroptosis through direct engagement of the NLRP3 inflammasome, in addition to DNA-recognition receptors such as TLR9 [18]. Recently we demonstrated the diagnostic importance of ox-mtDNA release in MDS by showing a profound elevation of this DAMP in the PB of MDS specimens and in MDS murine models [19], suggesting that DAMPs such as ox-mtDNA aid in the maintenance of the evolutionary pressures that give rise to the malignant clone. Therefore, a better mechanistic understanding of the role of ox-mtDNA in MDS pathogenesis warrants further investigation.

Our previous studies also demonstrated the importance of the activation of the NLRP3 inflammasome in the initiation and development of MDS pathogenesis [5]. This work has led to understanding that the inflammatory microenvironment’s activation of pyroptosis leads to the phenotypic anemia in the disease. The NLRP3 protein, after an inflammatory stimulus, multimerizes, leading to the recruitment of the adaptor protein apoptosis-associated speck-like protein containing a CARD (PYCARD, ASC), which also polymerizes, creating ‘specks’ which can be used as predictors of MDS [20,21]. This further demonstrates that the NLRP3-mediated pyroptosis in MDS is not only a biproduct of disease pathogenesis but is linked to the development and evolution of disease progression. Due to the importance of this process in MDS, we investigated whether the pyroptotically released DAMP ox-mtDNA acts as a catalyst of inflammasome activation to perpetuate BM failure in MDS. We found ox-mtDNA aids in maintaining the pyroptotic microenvironment, via engagement and re-localization of TLR9, from the endoplasmic reticulum (ER) to the cellular surface in MDS cells, allowing the formation of functionally active Myddosome (MyD88) signaling complex [22]. Importantly, specific therapeutic interruption of the TLR9/ox-mtDNA axis decreases pyroptosis and improves hematopoiesis, demonstrating a novel targetable axis in MDS, a disease with few therapeutic options.

## 2. Results

### 2.1. Extracellular Ox-mtDNA Is a DAMP That Triggers Inflammasome Activation

Previous studies demonstrate that ox-mtDNA triggers sterile inflammation and has been implicated as an indispensable effector of NLRP3 inflammasome activation [13,14,23,24], and we reported that the pyroptosis-induced extracellular ox-mtDNA can serve as an MDS diagnostic marker [19]. We validated the increased levels of ox-mtDNA in lower risk (LR) disease, both in PB and BM plasma compared to healthy BM plasma and found increased levels of ox-mtDNA in both (Figure 1A). To evaluate its role as a non-canonical extracellular DAMP in pyroptosis, we treated SKM1 or U937 cells with increasing doses of synthetic ox-mtDNA (5, 50, and 500 ng/mL, Appendix A) and observed increased phosphorylation of NFκB (p65) and cleavage of caspase-1 at 50 ng/mL, indicating the induction of inflammasome assembly (Figure 1B).

While ox-mtDNA also activated IL-1β, there was no dose response since the lower band (17 kDa) was visible among all doses (Appendix A). The 50 ng/mL dosage is comparable to the average LR BM ox-mtDNA observed in Figure 1A, demonstrating that this level is physiologically relevant. Inflammasome activation was confirmed through the activation of IL-1β at 50 ng/mL ox-mtDNA (Figure 1C) and two-fold increase in caspase-1 activity (Figure 1D). Additionally, ox-mtDNA treatment resulted in lactate dehydrogenase (LDH) release indicating lytic cell death (Figure 1E). To validate the role of the NLRP3 inflammasome in ox-mtDNA pyroptosis activation, we either CRISPR knocked out (KO) NLRP3 with a pool of guides or treated cells with MCC950, an established NLRP3 inhibitor [5]. Caspase-1 activation by ox-mtDNA was abrogated by blockage of the NLRP3 inflammasome, confirming its role in our observations (Figure 1F). Similar results were observed with U937 cells (Appendix A). ASC specks are polymerized during inflammasome activation, diagnostically relevant [20], serve as a platform for caspase-1 binding and are also released upon lytic death [5,20,25]. We assessed whether ox-mtDNA induce ASC specks and observed a significant increase in ASC speck formation by immunofluorescence (IF) (Figure 1G,H), which was confirmed by Western blot demonstrating increased oligomerization and decreased monomer subunits in the treated cells (Figure 1I). Finally, to demonstrate ox-mtDNA treatment results in activation of pyroptosis, as opposed to apoptosis, we probed for PARP and caspase-3 activation, which was absent upon treatment (Figure 1J).

Having confirmed that MDS BM plasma has significantly high levels of ox-mtDNA [5,19] and that this excess is sufficient to induce pyroptosis in leukemic cell lines, we assessed if ox-mtDNA can also induce pyroptosis in healthy BM-MNCs. As expected, treatment with 50 ug/mL ox-mtDNA induced a significant caspase-1 activation and lytic cell death (LDH) in healthy BM (Figure 1K), and significantly decreased hematopoiesis evidenced by decreased colonies (Figure 1L,M). This confirms that ox-mtDNA is a DAMP capable of triggering the NLRP3 inflammasome, in a time- and dose-dependent manner, in cell lines and primary cells affecting hematopoietic potential.

### 2.2. MDS HSPC and Leukemic Cell Lines Have Increased Expression of TLR9

The observation that ox-mtDNA can induce direct effects on hematopoiesis indicates recognition by the target cells. An inflammatory receptor of cell free DNA is TLR9, which has also been shown to play a role in pyroptosis activation [26]. We observed abundant co-localization and increased expression of cytosolic ox-mtDNA and TLR9 in MDS HSPCs, compared to healthy BM-MNCs (Figure 2A, Appendix A). This colocalization was quantifiable with MDS cells having an average of 92% of ox-mtDNA bound to TLR9 compared to 72% in normal BM-MNCs (Figure 2B) with cytosolic concentration of ox-mtDNA, and TLR9 surface expression, significantly increased in the cytoplasm of MDS BM-MNCs, compared to healthy controls (Figure 2C). Moreover, there was a comparable 2.5-fold increase in TLR9 gene expression in MDS HSPCs (Figure 2D), confirming previous results showcasing higher TLR9 gene expression in MDS [27]. There was an increase in the percentage of TLR9^+^ cells, particularly CD34^+^TLR9^+^ cells, in MDS compared to normal BM (Figure 2E, Appendix A), and particularly in the stem (CD34^+^CD71^−^CD14^−^, CD34^+^CD33^−^) and progenitor (CD34^+^CD38^−^) populations (Figure 2F). While the overall common myeloid progenitor (CMP, CD34^+^CD38^−^) population was not significantly different between MDS and healthy specimens within this small sample size, stratification by risk showed that LR samples were significantly higher than healthy and HR samples (Figure 2G), similar to our observations that circulating ox-mtDNA is higher in lower risk specimens [19] and those of others showing TLR9 reduction during progression to transformation [27]. Moreover, tSNE analysis of TLR9^+^CD34^+^ cells showed both an increase in the patient population and, importantly, in their expression of myeloid markers denoted by CD14^+^ and CD33^+^ denoting myeloid skewing (Figure 2H, Appendix A).

TLR9 circulates through the cytoplasm to the surface prior to entering lysosomes, where it’s proximity with MyD88 directs activation [28]. We observed a strong lysosome induction, as read by LC3 and Lysotracker^®^ Deep Red, in MDS (Figure 2I,J). This increased lysosome activation in the MDS HSPC suggests that the TLR9 pathway might be triggered and functional in this disease. We further linked this phenomenon to excess ox-mtDNA in the plasma of MDS patients by demonstrating that treatment of normal BM-MNCs with ox-mtDNA results in increased lysosomes with internalized ox-mtDNA and TLR9 phenocopying MDS (Figure 2K).

### 2.3. TLR9 Is Necessary for Ox-mtDNA Directed Pyroptosis

Next, we investigated the impact of ox-mtDNA in the activation of the TLR9/IRF7 signaling axis. Upon incubation of SKM1 and U937 cells with ox-mtDNA, we observed a time-dependent internalization of ox-mtDNA and co-localization with TLR9 (Figure 3A). To confirm the binding of ox-mtDNA with TLR9, we immunoprecipitated TLR9 and confirmed its binding to oxDNA by immunoblotting and vice versa (Figure 3B). To further characterize the effect of TLR9 in the ox-mtDNA-mediated activation, we assessed NLRP3 inflammasome activation by ox-mtDNA, by caspase-1 cleavage, relative to the density of TLR9 expression: high TLR9 expression (SKM1), and medium (U937) and low to absent expression (THP1) (Figure 3C). Receptor density is an important determinant of the time interval to caspase-1 cleavage, with SKM1 cells responding within 1 h of treatment, U937 cells responding within 2 h, and THP1 cells showing no caspase-1 cleavage in response to ox-mtDNA exposure up to 4 h after ox-mtDNA incubation (Figure 3D). Importantly, after 4 h in SKM1 cells, most of the caspase-1 bands are depleted, compared to U937 and THP1, a mechanism used to restrict excessive inflammation [29]. Activation of IL-1β was also time-dependent and a later response in both types of cells, while still TLR9 density dependent, 4 h in SKM1 cells compared to 24h in U937 cells (Appendix A). Additionally, TLR9 KO cells showed observed an abrogation of IL-1β activation induced by ox-mtDNA (Appendix A).

To determine if TLR9 is responsible for NLRP3 inflammasome activation after ox-mtDNA exposure, we created a TLR9 knockout in SKM1 (high TLR9) and U937 (medium TLR9) cells using CRISPR/Cas 9 gene editing (Appendix A). Ox-mtDNA treatment induces phosphorylation of NFκB, and the maturation of caspase-1 and IL-1β (Figure 3E and Appendix A), as well as caspase-1 and LDH, in the scrambled control but not in TLR9 CRISPR KO SKM1 cells (Figure 3F,G, Appendix A) indicating that TLR9 is indispensable for ox-mtDNA-dependent inflammasome activation. To confirm the necessity of TLR9 to sensitize bystander cells to ox-mtDNA, we overexpressed TLR9 in THP1 (no TLR9) with a lentiviral vector prior to treatment with ox-mtDNA, which restored sensitivity to ox-mtDNA (Figure 3H,I). To assess if ox-mtDNA is directly being internalized by TLR9 via its lysosomal trafficking, we assessed ox-mtDNA-treated cells by LC3 and Lysotracker^®^ Deep Red co-localization of lysosome and oxDNA (Figure 3J, Appendix A). These data strengthen the conclusion that TLR9 is a main receptor for ox-mtDNA dependent inflammasome activation, and the significant plasma membrane translocation of TLR9 upon exposure to ox-mtDNA (Figure 3K, Appendix A) indicates that ox-mtDNA leads into a feed-forward loop of pyroptosis.

### 2.4. IRF7 Signaling Is Activated by Ox-mtDNA/TLR9 Engagement

To confirm TLR9 activation by ox-mtDNA, we tested potential downstream mediators (TBK1, IRF7, IRF3, and NF-κB) and found that IRF7 was activated by ox-mtDNA treatment of SKM1 cells (Figure 4A), including nuclear translocation upon ox-mtDNA treatment, corroborating its activation (Figure 4B,C). This IRF7 activation occurred rapidly (nuclear translocation within 30 min, Figure 4D, Appendix A) and required TLR9 expression, as TLR9 CRISPR KO prevented ASC formation and IRF7 nuclear translocation (Figure 4E,F, Appendix A). Moreover, this activation correlated with the level of TLR9, as SKM1 cells had a faster IRF7 translocation (30 min) compared to U937 cells (4 h), which have comparatively less TLR9 (Appendix A, Figure 3D,F). Lastly, taking advantage of our previously published RNA-seq dataset [30] comparing healthy versus LR MDS specimens, we found increased TLR9 pathway activation, including of CTSB and IRF7, which were accordingly activated by ox-mtDNA (Appendix A).

At baseline, MDS patients have significantly higher levels of type 1 interferons (IFNs) [31] and, accordingly, gene expression of the interferon-stimulated genes (ISGs) IFNα1, IFNα10, IFNβ1, CXCL10, ISG15, SAMD9L, and IFI27L2 were elevated in MDS in data obtained from 213 WHO-defined MDS patient specimens at time of diagnosis, as well as from 20 healthy donors from the National Taiwan University Hospital (Figure 4G, Appendix A; additionally, IFNα2, α4, α5, α8, α14, and α21 all significantly elevated). This activation correlated with increased expression of TLR9-mediators, including IRF7 and IRAK in MDS, although there was no difference with regard to risk (Figure 4H, Appendix A). To confirm that the ox-mtDNA is involved in ISG induction through the TLR9/IRF7 axis, we analyzed the gene expression of type I IFN genes after treatment with ox-mtDNA and found a time-dependent increase in their activation (Figure 4I, Appendix A). This activation was mediated through TLR9 activation, as silencing this receptor abrogated increased ISG expression after ox-mtDNA treatment (Figure 4J, Appendix A). These findings show that ox-mtDNA engages and activates TLR9 through IRF7 nuclear translocation.

### 2.5. Ox-mtDNA/TLR9 Signaling Can Be Therapeutically Targeted in MDS

Recently, studies have shown that mtDNA/TLR9 ligation is linked to anemia development during inflammation [32]. Having established that MDS BM plasma is sufficient to induce pyroptosis in normal BM-MNCs and that MDS BM plasma has significantly high levels of ox-mtDNA [5,19], we assessed the specific impact of ox-mtDNA and TLR9 on hematopoietic potential. Primary healthy BM-MNCs transfected with lentivirus vectors to overexpress (OE) or knock out (KO) TLR9 with a pool of specific CRISPR guides (Appendix A) were treated with synthetic ox-mtDNA and their colony-forming capacity was assessed. As expected, ox-mtDNA treatment of healthy BM-MNCs significantly reduced colony-forming capacity (Figure 5A,B). TLR9 OE increased sensitization of healthy BM-MNCs to ox-mtDNA, particularly in the erythroid compartment (BFU-E, Figure 5A,B), while TLR9 KO disrupted the effect of ox-mtDNA with hematopoiesis matching untreated levels (Figure 5A,B). These data indicate that ox-mtDNA accumulates in the microenvironment in MDS where it binds TLR9 sensitizing cells to inflammasome activation affecting hematopoiesis.

Next, we evaluated the therapeutic potential of targeting the ox-mtDNA/TLR9 axis by testing the ability of several compounds to block caspase-1 activation in SKM1 or U937 cells (Figure 5C and Appendix A). To test the role of cGAS, another nucleic acid sensor that we have recently shown to recognize intracellular DAMPs, we used a non-TLR9 targeting cGAS inhibitor (RU.521) or CRISPR KO cGAS cells (Figure 5C). We used TLR9 KO cells as a negative control. We found that only TLR9 KO cells were able to significantly reduce ox-mtDNA activation of caspase-1, but not cGAS inhibition or KO. To validate the therapeutic potential of these inhibitors in MDS, we tested hematopoietic potential in primary MDS BM-MNCs. Caspase-1 activation was prevented by either blocking TLR9 signaling with an IRAK inhibitor that prevents downstream activation of Myd88, oligodeoxynucleotide (ODN)-F (TLR9 antagonist), blocking ox-DNA with IRS954 (an inert non-sense ODN), preventing lysosome internalization with HCQ, the inflammasome inhibitor MCC950, or depleting excess ox-mtDNA with a soluble TLR9-IgG4 chimeric molecule (the ectodomain of TLR9 fused to the Fc domain of human IgG4) developed by us to serve as a decoy receptor or ligand trap (Appendix A). Blocking TLR9 signaling with IRAKi, trapping excess ox-mtDNA with TLR9 chimera, blocking binding to TLR9 with ODN-F, or preventing lysosomal internalization significantly improved the colony-forming capacity of MDS BM-MNCs (Figure 5D), confirming the therapeutic potential of targeting the ox-mtDNA/TLR9 axis in MDS. 

## 3. Discussion

Ox-mtDNA is a novel diagnostic biomarker [19] that may represent a therapeutic target for MDS. Here, we demonstrate the role that the diagnostically evident excess of ox-mtDNA plays in hematopoietic potential by inducing the overexpression and engagement of TLR9. Our findings identify it as a key DAMP contributing to both medullary HSPC pyroptosis and propagation of sterile inflammation in MDS. We show that incubation with ox-mtDNA provides a secondary signal that is sufficient to induce activation of the canonical NLRP3 inflammasome pathway and pyroptotic cell death, as evidenced by cleavage of caspase-1 and IL-1β, and ASC speck formation and release. Our studies also suggest that strategies that effectively neutralize extracellular ox-mtDNA/TLR9 may suppress DNA-sensor-directed inflammation in the BM niche and possibly improve hematopoiesis. Indeed, strategies that mitigate mitochondrial membrane depolarization, through activation of the Nrf2-antioxidant pathway, or the binding of ox-mtDNA to its cognate TLR9 receptor, may offer promising therapeutic potential [33,34]. Moreover, these findings may be extended to other disorders in which ox-mtDNA has been implicated in innate immune activation [14,35,36,37].

The important role that pyroptosis plays in the development of the phenotypic cytopenias of MDS demonstrates the pathogenetic involvement of the innate immune microenvironment which provides the selective pressure for evolution of malignant/defective cells [4,5]. The role of extracellular ox-mtDNA in this process is supported by TLR9’s reduction after transformation, when the biological pressure for survival and evolution is no longer needed [27]. Our investigation demonstrates that TLR9 translocates to the plasma membrane in MDS HSPCs, where it can engage nucleic acid-based DAMPs, such as ox-mtDNA, further enhancing pyroptosis and reinforcing surface TLR9 translocation. In this way, we expect that ox-mtDNA contributes to accelerating the death of healthy HSPC and selection of increasingly aggressive malignant clones. Hence, another potential consequence of ox-mtDNA/TLR9 targeting is to therapeutically prevent the evolution towards leukemia.

Early in the disease progression, abundance of ox-mtDNA in MDS reinforces surface TLR9 translocation, initiates transcription of ISGs, inflammatory cytokines, and further induction of inflammasome activation. Pellagati et al. demonstrated that ISG transcription is the most upregulated pathway in MDS [31]; our research suggests that this could be a result of ox-mtDNA/TLR9 pathway activation. Inflammasome activation then results in both cell death and proliferation via β-catenin activation [5,38] and IL-1β, which has been implicated in immuno-senescence and myeloid skewing with aging (reviewed in [39]). This activation also results in degradation of the erythroid transcription factor GATA1, through caspase-1 activation [40], changing the ratio between GATA1 and the myeloid transcription factor PU.1 favoring myeloid commitment, maturation arrest, and anemia [40,41]. Additionally, chronic TLR activation in HSPC causes loss of quiescence with recruitment into the cell cycle and HSPC depletion [42,43,44,45]. Having established the importance of ox-mtDNA/TLR9 in MDS, a next step will be to understand its contribution to the phenotype: ISG overexpression, myeloid skewing, immune-senescence, and BM failure.

Recent publications show a novel role for red blood cells (RBCs) as an anti-inflammatory sink for mtDNA [32,46,47,48]. MDS is especially sensitive to the loss of this natural ox-mtDNA removal system due to HSPC pyroptosis, which contributes to both the inflammatory milieu and cytopenias. In this manner, the loss of RBCs, coupled with increased ox-mtDNA, results in further HSPC death and feed-forward BM failure. Importantly, we also demonstrate that TLR9 is indispensable for ox-mtDNA-initiated inflammasome activation proportionate to cellular TLR9 density, highlighting the viability of therapeutically targeting this ligand/receptor interaction. We showcase that TLR9 agonists, or a chimeric TLR9-IgG trap we developed, improved hematopoietic potential in MDS BM explants through the reduction in inflammasome activity. Despite the role of cGAS as another potential receptor for ox-mtDNA, our work demonstrates that inhibiting this pathway did not affect the signaling studied here. However, considering pyroptotic release of other nucleic acid-based DAMPs, it will be important to dissect their contribution to this phenotype to assess the hierarchy of this axis in MDS pathogenesis. However, our data demonstrate that TLR9 is critical, so the use of a soluble TLR9 trap could be beneficial at clearing other potential DAMPs that use TLR9 for signaling.

## 4. Materials and Methods

### 4.1. Patient Samples

Normal samples were obtained from Stem Express (Folsom, CA, USA). MDS specimens were acquired from consented MDS patient specimens through Moffitt’s Total Cancer Care™ Protocol [49]. De-identified tissues were released for this study, stratified according to the International Prognostic Scoring System (IPSS). The study was conducted in accordance with the Declaration of Helsinki and approved by the Institutional Review Board of the University of South Florida.

### 4.2. Cells

THP1 (TIB-202), HEL 92.1.7 (TIB-180), and U937 (CRL-1593.2) cells were obtained from American Type Culture Collection (ATCC) and SKM1 (ACC 547) from the Leibniz-Institute DSMZ–German Collection of Human & Animal Cell Lines. THP1 and HEL overexpressing TLR9 were transduced via lentiviral infection, as previously described, with pcDNA3-TLR9-YFP, which was a gift from Doug Golenbock (Addgene plasmid # 13642) and selected with Neomycin 3 days post-transfection.

### 4.3. Colony Forming Capacity

MDS BM mononuclear cells (BM-MNCs), treated as described in the results, were plated in duplicate (1 × 10^5^ cells/dish) in MethoCult™ (H4434, StemCell Technologies, Vancouver, BC, Canada) as previously described [5,20]. After incubating for 14 days, colonies were counted using the StemVision microscope and software (Catalog # 22006, StemCell Technologies) and counts were validated manually.

### 4.4. CRISPR

TLR9-deficient cells were created using CRISPR CRIPSR/Cas9 gene editing with RNA guides TLR9 (F- CACCGTTGCAGTTCACCAGGCCGTG R- AAACCACGGCCTGGTGAACTGCAAC), or Scrambled control (F- GACGGAGGCTAAGCGTCGCA, R- TGCGACGCTTAGCCTCCGTC) into a puromycin resistance pL-CRISPR.SFFV plasmid (a gift from Benjamin Ebert, Addgene plasmid # 57829) [22]. Forward and reverse guide oligonucleotides were purchased from Integrated DNA Technologies (Coralville, IA, USA). CACC on forward and AAAC on reverse oligonucleotides were added 5′ for plasmid ligation. CRISPR plasmids were packaged into lentivirus and transduced as previously described [5].

### 4.5. Inflammasome Activation

Ox-mtDNA for treatment was synthesized from mtDNA extracted using the Mitochondrial Extraction Kit according to the manufacture’s protocol (Active Motif, Carlsbad CA, USA) and amplified by ND1 primers (ND1 Forward: 5′-CCCTAAAACCCGCCACATCT-3′; ND1 Reverse: 5′-GAGCGATGGTGAGAGCTAAGGT-3′) with the addition of oxidized guanosine to the master mix and mtDNA amplified, as described in [13] (Appendix A). The pyroptotic TLR4 signaling pathway was activated by incubation with LPS, ATP, and nigericin (LAN) [23]. Caspase-Glo^®^ 1 Inflammasome and LDH-Glo^®^ Cytotoxicity assays (Promega Corporation, Madison, WI, USA) were used to assess inflammasome activation following manufacturer’s protocols. Ox-mtDNA levels were quantified using the DNA/RNA Oxidative Damage (High Sensitivity) ELISA Kit (Cayman Chemical Company, Ann Arbor, MI, USA) [19].

### 4.6. Inhibitors

Cells were treated with inhibitors for 1 h prior to the addition of ox-mtDNA. Inhibitors used: our developed TLR9-IgG4 chimera at 50 ng/ul, and corresponding isotype; 0.5 uM IRAK 1/4 inhibitor (Caymen 17540, CAS 509093-47-4), 1 uM ODN 4048-F (TLR9 antagonist) and ODN 2395 control (Invivogen, San Diego, CA, USA), 30 uM Hydroxychloroquine (HCQ, Sigma-Aldrich, St. Louis, MO, USA) to accumulate autophagosomes for imaging and 10 uM for CFAs, 20 ug/mL IRS95424, 1 µM RU.521 (cGAS inhibitor, Aobious, Gloucester, MA, USA).

### 4.7. Immunofluorescence

Immunofluorescent staining was performed as undertaken previously [5] and stained with conjugated primary antibodies. Nuclei were stained with ProLong^®^ Gold anti-fade reagent with DAPI (Thermo Fisher Scientific, Waltham, MA, USA). Primary antibodies used were anti-oxDNA/RNA FITC (Abcam), CD289 (TLR9) APC (Thermo Fisher Scientific), IRF7 Alexa Fluor 647 (Thermo Fisher Scientific). Lysotracker Deep Red (Invitrogen) staining was performed following the manufacturer’s protocol, with the addition of 30 uM HCQ (Sigma) to ensure accumulation of autophagosomes for imaging. Slides were imaged with a Leica SP8 laser scanning confocal microscope (Leica Microsystems GmbH, Wetzlar, Germany). Images were captured through a 63×/1.4 NA objective lens. Images were analyzed with LAS X software version 3.1.5 (Leica Microsystems GmbH, Wetzlar, Germany).

Quantification ASC specks, stained with anti-ASC (Santa Cruz, 1:200) and Alexa 488 goat anti-mouse (1:400), were counted from at least 200 cells per group from the MFI on the TIF images with Definiens Tissue Studio version 4.7 (Definiens AG, Munich, Germany). Nucleus detection and cell growth algorithms were used to segment individual cells within each image.

### 4.8. Western Immunoblot

Following treatment, the harvested cells were lysed in RIPA buffer with phosphatase and protease inhibitors as previously described [5]. Immunoprecipitations from specimen plasma were isolated using Protein A and G Agarose Fast Flow beads (Millipore Sigma, Burlington, MA, USA) according to the manufacturer’s protocol and using 4 µg of anti-ox-DNA antibody. The following antibodies were used: caspase-1, NF-κB p65, LC3, TLR9, Histone H3, caspase-3, PARP, IRF7, phospho-TBK1, phospho-IRF7, phospho-IRF3 (Cell Signaling Technology, Inc., Danvers, MA, USA), Human IL-1 beta/IL-1F2 (R&D Systems, Inc., Minneapolis, MN, USA), anti-oxDNA/RNA (Abcam), Anti-NLRP3 Antibody (Millipore Sigma), ASC (Santa Cruz), β-actin (Sigma-Aldrich), and appropriate Amersham ECL HRP Conjugated Antibodies (Thomas Scientific, Swedesboro, NJ, USA). Apoptosis positive control was A431 Whole Cell Lysate EGF Stimulated (Rockland Immunochemicals, Inc., Limerick, PA, USA).

To assess ASC oligomerization, cell pellets were harvested and lysed. The pellet was then resuspended in PBS and fresh 2 mM Disuccinimidyl suberate (DSS, Thermo Scientific), and incubated on a rotator for 30 min at room temperature. Following this DSS-crosslinking, the resultant was again pelleted for 10 m at 5000 rpm at 4 °C prior to blotting.

### 4.9. Flow Cytometry

Cryopreserved normal and MDS BM-MNCs were thawed, washed with 2% BSA-PBS, blocked with Human FcR Blocking reagent (Miltenyi Biotec; Bergisch Gladbach, Germany), stained for CD14, CD33, CD34, CD38, (Becton, Dickinson and Company, Franklin Lakes, NJ, USA), TLR9/CD289, CD71 (Thermo Fisher Scientific; Waltham, MA, USA), resuspended in 0.1 uM DAPI (Thermo Fisher Scientific), and run on the BD LSRII (Becton, Dickinson and Company) at Moffitt’s Flow Cytometry core facility. FCS files were analyzed using FlowJo v10 (FlowJo LLC. Ashland, Oregon).

### 4.10. Gene Expression Analysis

RNA extraction was performed with Qiagen RNeasy Mini Kit (Hilden, Germany) following the manufacturer protocol. RNA concentration and integrity were verified using the ND-1000 spectrophotometer (NanoDrop Technologies, Wilmington, DE, USA). RNA was reverse transcribed using qScript XLT cDNA Supermix (QuantaBio, Beverly, MA, USA) according to the manufacturer’s protocol. INFB1/Infb1 and GATA1 expression was quantified using TaqMan Advanced Gene Expression Assays, TaqMan Fast Advanced Mastermix, and 7900 HT Fast Real-Time PCR System (Life Technologies, Foster City, CA, USA). The rest were amplified using the primers in Appendix A with SYBR Green PCR Master Mix (Life Technologies). Data were analyzed using the ΔΔCt methodology with ACTB/Actb expression for TaqMan assays and GAPDH/Gapdh for SYBR assays as internal control.

Gene expression profiling data were also obtained from 213 WHO-defined MDS patient specimens at time of diagnosis, as well as from 20 healthy donors from the National Taiwan University Hospital using the Human HT-12 v4 Expression BeadChip (Illumina, San Diego, CA, USA) [30]. For each sample, 1.5 μg cDNA was hybridized to Illumina HumanHT-12 v4 Expression BeadChip according to the manufacturer’s instructions. Intensities of bead fluorescence were detected by Illumina BeadArray Reader, and the results were transformed to numeric values using GenomeStudio v2010.1 Software (Illumina).

### 4.11. Sandwich ELISA

The binding efficacy of the TLR9-IgG chimera was assessed using a Bethyl Labs (Montgomery, TX, USA) Sandwich ELISA kit (ELISA Starter Accessory Kit I), following the manufacture’s protocol for Indirect ELISA. Briefly, a 96-well plate was coated with Coating Buffer and TLR9-IgG chimera or IgG4 was added in a 2-fold dilution curve and, after blocking, the ODN 2006 Biotin (Invivogen) was added followed by streptavidin-HRP (Cell Signaling Technology). Binding of the chimera to CpG was assessed by colorimetric quantification of TMB substrate development.

### 4.12. Statistical Analysis

Statistical analyses were performed using GraphPad Prism Software: two treatment groups by unpaired Student t tests; fold change by paired Student t tests; and one-way ANOVA was applied to other data with multiple comparison analysis. All analyses and graphics show standard error of the mean bars (SEM) and *p* values < 0.05 are statistically significant. *p* values are shown as asterisk: * *p* ≤ 0.05, ** *p* ≤ 0.01, *** *p* ≤ 0.001, **** *p* ≤ 0.0001.

## Figures and Tables

**Figure 1 ijms-24-03896-f001:**
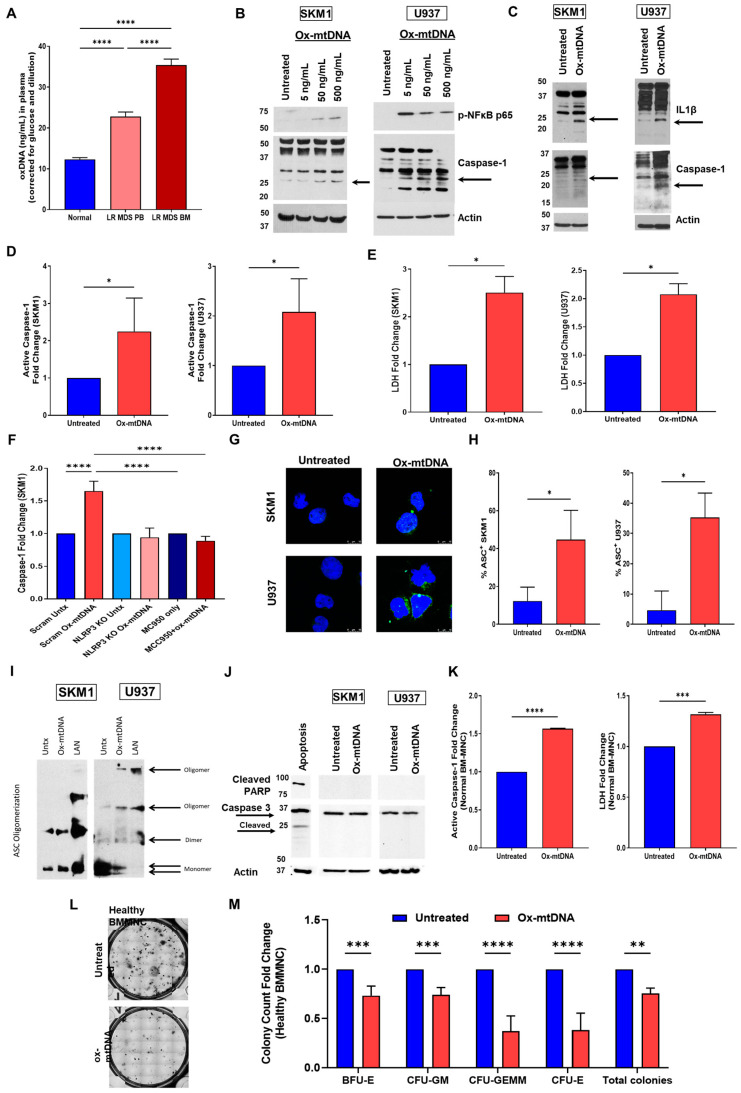
Ox-mtDNA activates pyroptosis. (**A**) Log10-transformed, glucose-adjusted ox-mtDNA levels from Low Risk (LR) MDS PB (n = 100) and BM plasma (n = 70), and Normal PB (n = 30). (**B**) Western blot of U937 and SKM1 cells treated with 10-fold increasing isolated mtDNA of the ND1 gene region amplified with oxidized guanosine (ox-mtDNA) for 2 h to induce Caspase-1 (arrow depicts cleaved fragment) and phosphorylated NFκB to establish dosage (representative blot of n = 3). Remaining figures, ox-mtDNA treatment is 50 ng/mL ox-mtDNA for 2 h unless otherwise stated. (**C**) Western blot of SKM1 and U937 cells treated with ox-mtDNA showing, cleavage of caspase-1 (arrow depicts cleaved fragment), and IL-1β (arrow depicts cleaved fragment) demonstrating inflammasome activation (representative blot of n = 3). (**D**) Fold change of caspase-1 activity quantified by Caspase-1 Glo^®^ assay in SKM1 and U937 cells treated with ox-mtDNA, (mean ± SEM of n = 5). (**E**) Fold change of LDH media release (measurement of cell death), quantified by LDH-Glo™ Cytotoxicity Assay, of SKM1 and U937 cells treated with ox-mtDNA (mean ± SEM of n = 5). (**F**) Fold change of caspase-1 activity quantified by Caspase-1 Glo^®^ assay in SKM1 cells pretreated with either CRISPR KO (pooled guides) for NLRP3 or 10 uM MCC950 for 48 h prior to treatment with ox-mtDNA (mean ± SEM of n = 4). (**G**) Representative confocal IF micrographs showing increased ASC specks in ox-mtDNA stimulated cells compared to untreated controls [DAPI (blue), ASC (green) (×2520)]. (**H**) Quantification of 1-to-2 μm ASC speck IF, at least 200 cells counted per group, (mean ± SEM of n = 3). (**I**) Immunoblot of ASC following chemical crosslinking, cells treated with ox-mtDNA or positive control LAN (LPS + ATP + Nigericin), arrow indicates oligomers. (**J**) Western Blot for PARP and Caspase-3. Western blot of SKM1 and U937 cells treated with ox-mtDNA, and an apoptosis positive control (A431 EGF Stimulated) showing induction of PARP1, cleavage of caspase-3 (arrow depicts cleaved fragments, representative blots of n = 3). (**K**) Fold change of caspase-1 activity quantified by Caspase-1 Glo^®^ assay and LDH media release quantified by LDH-Glo™ Cytotoxicity Assay in primary Normal BM-MNC cells treated with ox-mtDNA (mean ± SEM of n = 3). (**L**) Colony formation assay for hematopoiesis in Healthy BM-MNCs treated with 50 ng/mL ox-mtDNA for 14 days (representative picture). (**M**) Quantification of various hematopoietic progenitor colonies in response to treatment with 50 ng/mL of ox-mtDNA for 14 days (mean ± SEM of n = 4). In this figure, significance was assessed by either paired *t*-test (comparison between two groups only) or ordinary one-way ANOVA with multiple comparison analysis in GraphPad Prism. *p* values are shown as asterisk: * *p* ≤ 0.05, ** *p* ≤ 0.01, *** *p* ≤ 0.001, **** *p* ≤ 0.0001.

**Figure 2 ijms-24-03896-f002:**
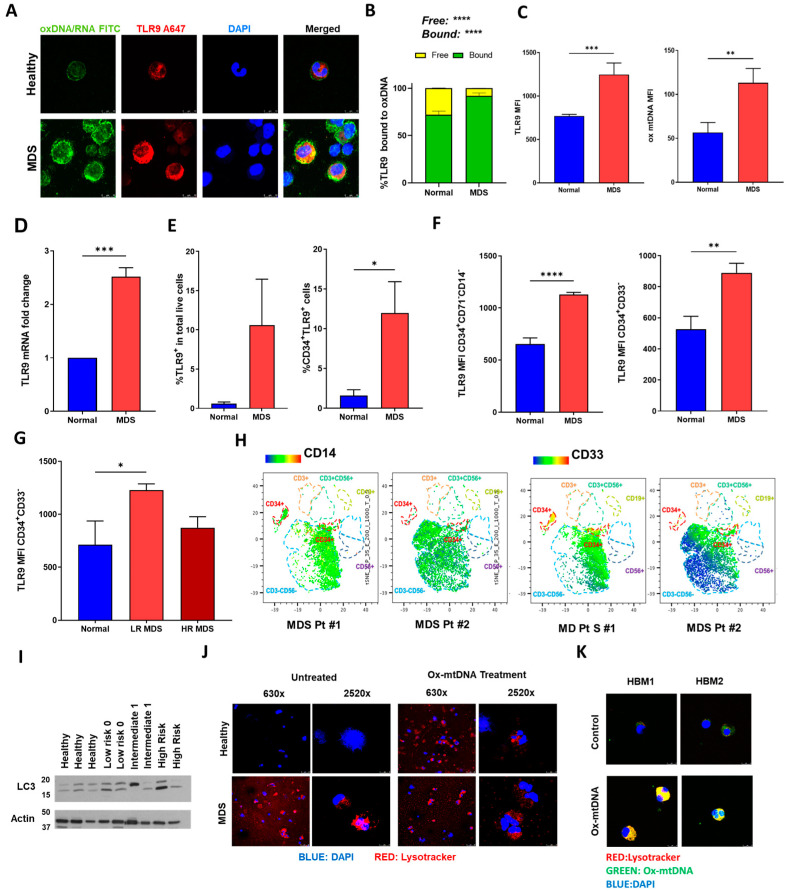
MDS HSPCs Display Surface TLR9 Induced by Cell-Free ox-mtDNA. (**A**) Confocal IF of colocalized TLR9 and ox-mtDNA in MDS BM-MNC samples compared with normal BM-MNC samples (×2520, DAPI, ox-mtDNA FITC, TLR9 A647). Representative Micrographs. (**B**) Mander’s Coefficient of Colocalization TLR9 bound to ox-mtDNA (Normal n = 3 and MDS n = 3). Significance measured by two-way ANOVA. (**C**) Quantification of mean fluorescence intensity (MFI) of TLR9 and oxDNA as in 2A, comparing Normal and LR MDS BM-MNCs (minimum 200 cells per sample, three samples per groups, mean MFI ± SEM). Significance measured by unpaired *t*-test. (**D**) Quantitative PCR to assess the relative expression of TLR9 mRNA (mean ± SEM. Normal n = 4, LR MDS n = 4). Significance measured by paired *t*-test. (**E**) Percent of cells expressing surface TLR9 measured by flow cytometry in either total live cells (left panel) or CD34^+^TLR9^+^ from all live cells (Normal = 4, MDS = 13). Significance measured by Welch’s *t*-test. (**F**) CD34^+^, CD71^−^, CD14^−^ cells, and CD34^+^ CD33^−^ (Normal = 3, MDS = 6). Significance measured by unpaired *t*-test. (**G**) Common Myeloid Progenitors, CD34^+^, CD38^−^ (Normal = 3, LR MDS = 3, HR MDS = 3). Significance measured by ordinary one-way ANOVA. (**H**) Flow cytometric tSNE analysis of TLR9 distribution in CD14^+^ and CD33^+^ populations (two representative MDS specimens are shown, an enlarged visualization of this figure is also in Appendix A). (**I**) Western blot of healthy and MDS BM-MNC lysates probed for the lysosomal marker LC3 and actin (3 Normal donors and 6 MDS donors with risk as shown). (**J**) Representative confocal IF micrographs showing increased lysotracker in MDS BM-MNCs compared to Healthy BM-MNCs; this increase is phenocopied in Normal BM-MNCs following treatment with 50 ng/mL ox-mtDNA for 2 h. [DAPI (blue), Lysotracker (Red)]. Three independent experiments. Nuclear MFI quantified in Appendix A). (**K**) Confocal IF of Healthy BM-MNCs, control and treatment with 50 ng/mL ox-mtDNA for 2 h, co-localization of lysosome and oxDNA demonstrated by yellow pixels (DAPI (blue), oxDNA (green), Lysotracker (Red)). Representative figures of 10 from 2 healthy donors are shown. *p* values are shown as asterisk: * *p* ≤ 0.05, ** *p* ≤ 0.01, *** *p* ≤ 0.001, **** *p* ≤ 0.0001.

**Figure 3 ijms-24-03896-f003:**
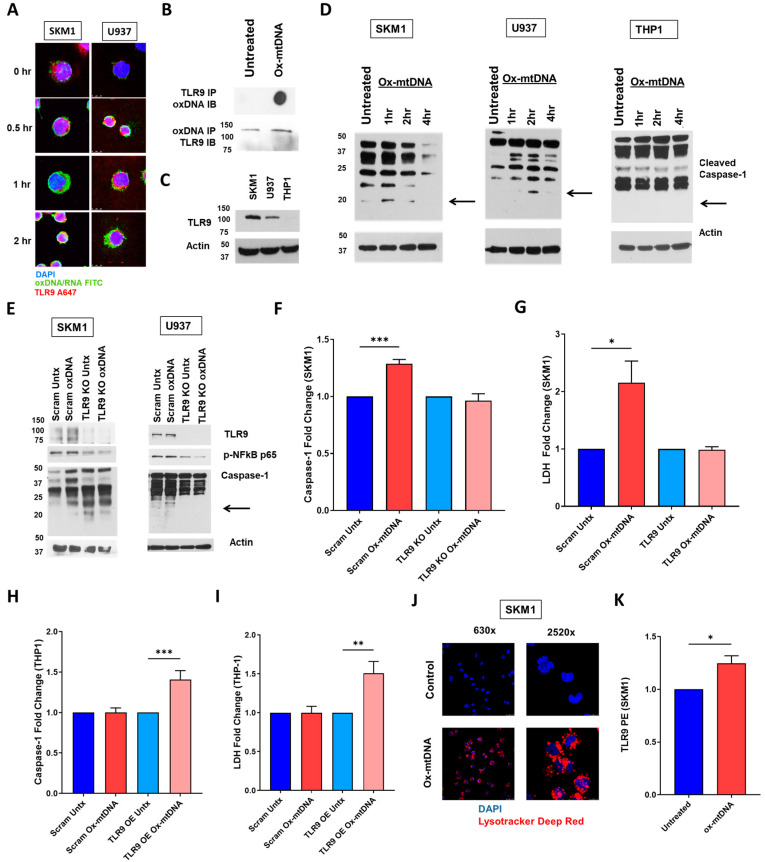
Ox-mtDNA triggers internalization via TLR9 activation. (**A**) Time-dependent co-localization of ox-mtDNA with TLR9 and receptor:ligand internalization demonstrated by confocal IF, in SKM1 and U937 cells [DAPI (blue), oxDNA (green), TLR9 (Red), representative micrographs of three independent experiments]. (**B**) Following ox-mtDNA treatment, TLR9 immunoprecipitated to assess the binding of TLR9 to oxDNA by dotblot and immunoblot (representative of n = 3). (**C**) Comparative TLR9 protein expression in SKM1, U937, and THP1 by Western blot (representative of n = 3). (**D**) SKM1, U937, or THP1 cells treated with 50 ng/mL ox-mtDNA followed by caspase-1 Western blot. (**E**) Western blot of caspase-1 and NFκB activation in SKM1 and U937 cells using CRISPR/Cas 9 gene editing to KO TLR9 expression. (**F**) Fold change of caspase-1 activity and (**G**) LDH media release in TLR9 KO SKM1 and U937 cells treated with ox-mtDNA (mean ± SEM of n = 3). (**H**) Fold change of caspase-1 activity and (**I**) LDH media release in TLR9 overexpression THP-1 cells treated with ox-mtDNA (mean ± SD of n = 3). In F through I, significance was assessed by ordinary one-way ANOVA with multiple comparison analysis. (**J**) Confocal IF of SKM1 cells treatment with 50 ng/mL ox-mtDNA for 2 h to assess induction of lysosomes [DAPI (blue), Lysotracker (Red)]. Representative micrographs, three independent experiments. (**K**) Change in surface expression of TLR9 in response to ox-mtDNA treatment by flow cytometry. Significance analyzed by Student *t*-test (MFI ± SD, n = 3)*. p* values are shown as asterisk: * *p* ≤ 0.05, ** *p* ≤ 0.01, *** *p* ≤ 0.001.

**Figure 4 ijms-24-03896-f004:**
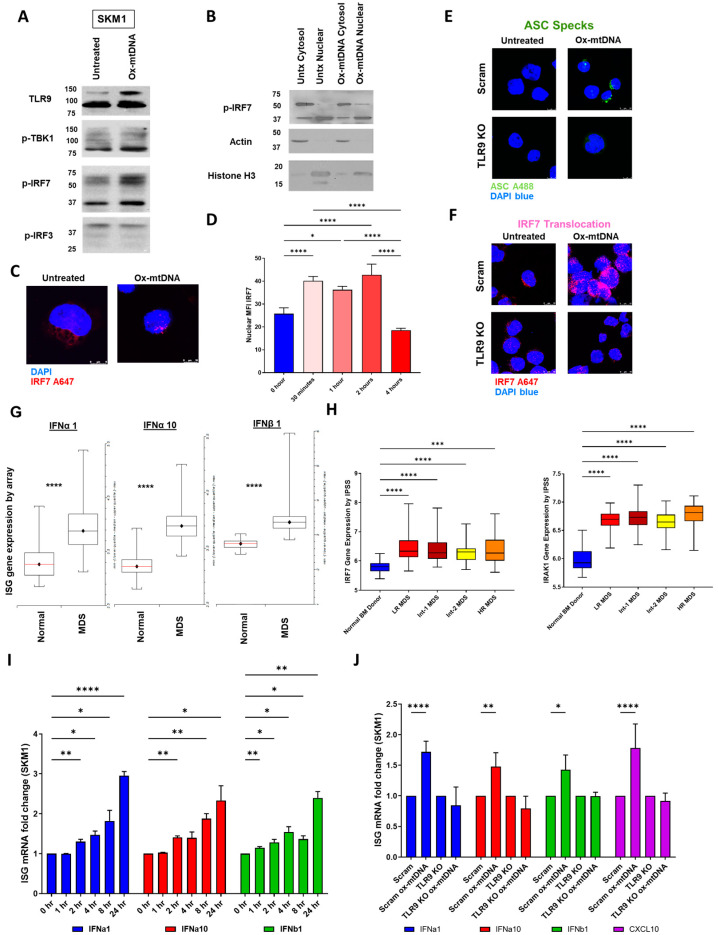
IRF7 signaling is activated by ox-mtDNA/TLR9 engagement. (**A**) TLR9 pathway activation demonstrated by TLR9 cleavage, and IRF7 phosphorylation. TBK1 and IRF3 unchanged suggesting other DNA sensing pathways are not activated. Representative Western Blot. (**B**) IRF7 translocation to the nucleus (cytosol denoted by presence of actin, nucleus denoted by presence of Histone H3. Representative Fractionation Western Blot. (**C**) Confocal IF to demonstrate IRF7 translocation, [DAPI (blue), IRF7 (Alexa Fluro 647) (×2520)]. (**D**) Quantification of nuclear MFI of IRF7 (translocation) of IF of SKM1 cells treated with ox-mtDNA at the times shown. Significance was assessed by ordinary one-way ANOVA. (**E**) Confocal IF of TLR9 KO on ASC speck formation and (**F**) IRF7 nuclear translocation [DAPI (blue), ASC (Alexa Fluro 488), IRF7 (Alexa Fluro 647) (×2520)]. (**G**) Gene expression array of Type 1 interferons IFNα1, α10, β1 (shown) although IFNα 1, 2, 4, 5, 8, 10, 14, and 21 were all significantly elevated (MDS n = 213, BMT donors n = 20). (**H**) IRF7 and Interleukin 1 Receptor Associated Kinase (IRAK)1 (n = 213, BMT donors n = 20). (**I**) Quantitative qPCR of IFNa1, IFNa10, IFNb1 gene expression in SKM1 cells treated with 50 ng/mL ox-mtDNA treatment 1–24 h (mean fold change ± SEM, n = 3). (**J**) Quantitative qPCR of IFNa1, IFNa10, IFNb1 and CXCL10 gene expression in SKM1 cells, with or without TLR9 CRISPR KO, treated with 50 ng/mL ox-mtDNA (mean fold change ± SEM, n = 3). *p* values are shown as asterisk: * *p* ≤ 0.05, ** *p* ≤ 0.01, *** *p* ≤ 0.001, **** *p* ≤ 0.0001.

**Figure 5 ijms-24-03896-f005:**
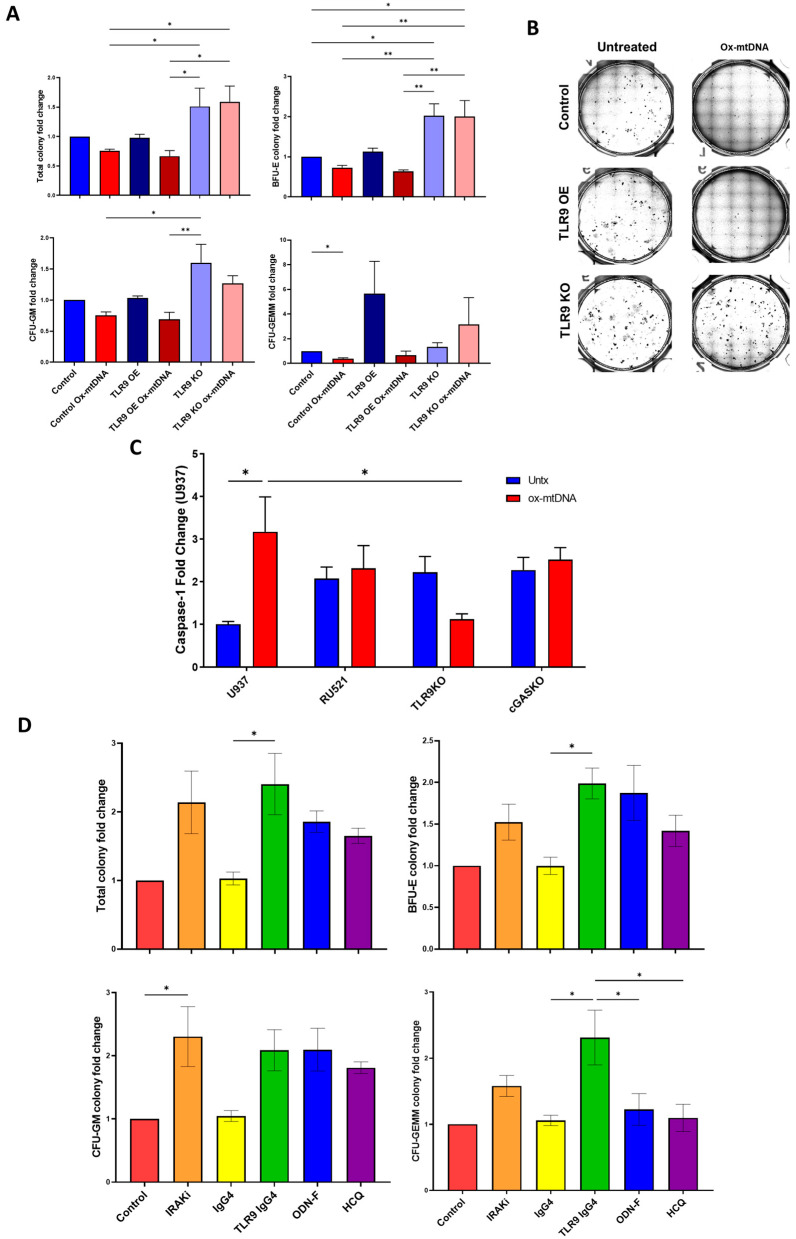
ox-mtDNA/TLR9 signaling as a therapeutic target to improve MDS. (**A**,**B**) Colony formation assay for hematopoiesis in Healthy BM-MNCs treated with 50 ng/mL ox-mtDNA for 14 days (mean ± SEM n = 3). Cells treated with lentivirus for TLR9 overexpression (OE), and TLR9 CRISPR knockout (KO). (**C**) Fold change of caspase-1 activity of U937 cells treated with cGAS inhibitor (RU.521) or CRISPR KO for TLR9 or cGAS followed by ox-mtDNA treatment (mean ± SEM of n = 3). (**D**) Colony formation assay for hematopoiesis in LR MDS BM-MNCs treated with inhibitors 14 days (mean ± SEM of n = 3). *p* values are shown as asterisk: * *p* ≤ 0.05, ** *p* ≤ 0.01.

## Data Availability

The data presented in this study are available on request from the corresponding author. The data are not publicly available due to restrictions with patient information.

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
