# Peer review of "Oxidized Mitochondrial DNA Engages TLR9 to Activate the NLRP3 Inflammasome in Myelodysplastic Syndromes"

_ijms, 2023, doi:10.3390/ijms24043896_

Round 1
Reviewer 1 Report
Building on their previous work, Ward and colleagues describe how ox-mtDNA acts through TLR9 to activate the inflammasome in a feed forward loop in Myelodysplastic Syndromes. The authors begin by showing that ox-mtDNA acts as a DAMP to activate pyroptosis. HSPC from MDS patients have increased TLR9 expression levels compared to controls, MDS cells also displayed increased lysosomal induction, which also occurred after ox-mtDNA treatment. Upon ox-mtDNA treatment, ox-mtDNA was internalised and found to interact with TLR9. Pyroptosis due to ox-mtDNA was found to be dependent on TLR9. IRF7 signalling was also activated by ox-mtDNA, dependent on TLR9, inducing type I interferons and interferon stimulated genes. Finally, the authors demonstrate that targeting the ox-mtDNA/TLR9 signalling axis may be of therapeutic benefit in MDS.
Major revisions
I have two major points that I believe need to be addressed before this manuscript is considered for publication.
Throughout the manuscript the authors state that the inflammasome activation they observe is due to NLRP3 activation, however they do not specifically look at the role of NLRP3 in this study, but instead rely on general indicators of inflammasome activity (caspase-1 and IL-1B cleavage, increased levels of LDH etc). Whilst I’m aware of the author’s previous work and the general literature around ox-mtDNA and NLRP3, without additional controls looking specifically at NLRP3 in the context of the current manuscript it is not appropriate to assume NLRP3 activation is leading to the observed inflammasome activation. Therefore, specifically testing the role of NLRP3 in this context is crucial and furthermore should be relatively straight forward, perhaps using NLRP3 genetic knockouts and/or inhibitors of NLRP3 as controls in the already presented experiments.
I do not agree with the authors statement that inhibition of cGAS does not affect caspase-1 activation (line 321); in regard to Figure 5C, it does appear the inhibition of cGAS by RU.521 reduces caspase-1 activity as compared to the vehicle control, albeit to a lesser extent than TLR9 inhibition. This apparent decrease should be further explored to determine if real or not, perhaps by using a STING inhibitor in addition to RU.521 or by genetic knockout of cGAS or STING. Alongside this, the role of cGAS/STING in the induction of type I interferons post ox-mtDNA treatment would also be interesting to look into. Given a cGAS/STING/NLRP3 signalling axis has been reported to be involved in MDS (McLemore et al 2022 JCI insight, McLemore et al 2018 Blood), further evidence is needed before stating cGAS is not involved in the signalling pathway described in this manuscript.
Minor comments
I also had some minor comments mainly regarding clarity of the figures and text.
-Some graphs are lacking a Y-axis label, please add appropriate labels.
-Graph axis size, border thickness of bar graphs, spacing between some images/graphs etc. are inconsistent throughout the figures, please keep consistent.
-Please add size markers for all western blot images.
-In Figure 1B and 3C please specify that the p-NFKB antibody refers to the p65 subunit.
-In Figure 1H for the SKM1 cells why is there is no difference in ASC oligomerisation in untreated and ox-mtDNA samples?
-Please keep the presentation of p values consistent throughout the figures, i.e. only use stars or numerical values.
-For experiments based to imaging (e.g. Figures 2J,K and 4C,D,E) please provide quantification of images.
-For Figure 4G is this graph showing gene expression? Please include Y axis and more detail in figure legend.
-Line 146-148: wording could be made clearer.
-Line 295: IRAK inhibitors are not direct inflammasome inhibitors so should not be referred to as such.
Author Response
Please see attached file with answers to both reviewers comments.

Reviewer 2 Report
Ward et al provide novel insights into inflammation associated with MDS with a focus on the role of TLR9 and NLRP3 in response to ox-mtDNA. The study is very well carried out, rigorously performed and methods well described. The results support the conclusions and are also sufficiently discussed in the context of the literature. Some of my suggestions to improve the manuscript are as follows:
1. Kindly comment on the role of NLRP3 in your studies in the introduction section (Lines 72-79).
2. Figure 1 needs to be improved. The role of NLRP3 in pyroptosis should be assessed using NLRP3 inhibitors such as MCC950. This is critical to demonstrate NLRP3 dependency in the author's experimental system. Also the statement made in line 141 needs to be supported experimentally.
3. Figure 1 - (a) Different fragments of Caspase-1, IL-1b and the ASC oligomers need to be indicated with their respective molecular weight. (b) Please confirm if the blots in 1C are in proper order. Caspase-1 activation blot for SKM1 and U937 looks drastically different than that shown in 1B for the same dose and time conditions. (c) It is worthwhile to show dose dependence with respect to IL-1b in conjunction with a ELISA readout.
4. There appears to be some misplacing of references in lines 145, 154. The references are not reflective of the statement. Also cite a proper reference for line 167. Sentence in line 170 should be restated as "might be triggered".
5. Figure 2- Figure 2D lacks tick marks for y axis. Statistics are not shown for 2E. Figure 2H is extremely difficult to read and needs to be enlarged with proper resolution.
6. Figure 3 - In 3D it is not clear why active Caspase-1 fragment disappears with time. SKM1 cells show reduced procaspase-1 at 4h however no active fragment is seen? Also is there a reason for Procaspase-1 levels not reducing in U937 cells although in figure 1B it is evident with 500ng/ml ox-mtDNA.
7. Figure 3E needs a better resolution blot. Is the increased p-NFKB in TLR9KO SKM1 cells an aberration as it is not seen with U937 cells?
8. Statement made in 239-242 with respect to figure 3J needs to be validated with TLR9 KO SKM1 cells which have been used in 3E.
9. Kindly elaborate on what the scores mean in figure S6C (lines 254-258)
10. Please include statistics for the following figures - 4G, 4H, 5A, 5D, S7B, S8A, S9D. Confirm if data with RU521 in figure 5C is significant.
Author Response
Attached file contains answers to both reviewer's comments.

Round 2
Reviewer 1 Report
The authors have thoroughly addressed all of my comments, I recommend the article to be accepted in present form.
Reviewer 2 Report
The manuscript has been sufficiently revised and all my concerns have been addressed.